# Inactivation of bacteria using synergistic hydrogen peroxide with split-dose nanosecond pulsed electric field exposures

**Zachary Rosenzweig**[1], **Jerrick Garcia**[1], **Gary L. Thompson**[2], **Lark J. Perez**[3]*

**1** Department of Chemical Engineering, Rowan University, Glassboro, New Jersey, United States of America, **2** WuXi AppTec, Philadelphia, Pennsylvania, United States of America, **3** Department of Chemistry and Biochemistry, Rowan University, Glassboro, New Jersey, United States of America

* perezla@rowan.edu

**Data Availability Statement:** All relevant data that support the findings of this study are available in the manuscript and Supporting information files.

## Abstract

The use of pulsed electric fields (PEF) as a nonthermal technology for the decontamination of foods is of growing interest. This study aimed to enhance the inactivation of *Escherichia coli*, *Listeria innocua*, and *Salmonella enterica* in Gomori buffer using a combination of nsPEF and hydrogen peroxide ($H_2O_2$). Three sub-MIC concentrations (0.1, 0.3, and 0.5%) of $H_2O_2$ and various contact times ranging from 5–45 min were tested. PEF exposures as both single (1000 pulse) and split-dose (500+500 pulse) trains were delivered via square-wave, monopolar, 600 ns pulses at 21 kV/cm and 10 Hz. We demonstrate that >5 log CFU/mL reduction can be attained from combination PEF/$H_2O_2$ treatments with a 15 min contact time for *E. coli* (0.1%) and a 30 min contact time for *L. innocua* and *S. enterica* (0.5%), despite ineffective results from either individual treatment alone. A 5 log reduction in microbial population is generally the lowest acceptable level in consideration of food safety and represents inactivation of 99.999% of bacteria. Split-dose PEF exposures enhance lethality for several tested conditions, indicating greater susceptibility to PEF after oxidative damage has occurred.

## Introduction

The decontamination of microbes in/on food products remains a global challenge [1]. Bacterial pathogens that are present can cause human illness or disease outbreaks resulting in significant financial and societal costs [2]. The most common strategies for decontamination of food products can be broadly classified as physical or chemical decontamination methods [3]. Physical methods for disinfection include treatment with heat, non-ionizing radiation, filtration, or electrical fields [4]. Chemical methods of disinfection involve treatment with a broad range of chemicals including alcohol, hypochlorite, hydrogen peroxide, and ozone. No single method exists for microbial decontamination of all food products, however, and the development of novel disinfection methods represents an important area of research and development. The application of non-thermal combination treatments for the disinfection of food products represents an especially promising area of research to preserve raw food

**Funding:** This work was funded by the USDA National Institute of Food and Agriculture's Agriculture and Food Research Initiative (NIFA AFRI) Food Safety and Defense Program (grant # 2022-67018-36540). The funders had no role in study design, data collection and analysis, decision to publish, or preparation of the manuscript.

**Competing interests:** The authors declare no competing interests.

characteristics, especially in cases for which synergistic effects are identified for the combination treatments [1, 5, 6].

Pulsed electric fields (PEFs) use short (typically μs-ms), high-voltage pulses that generate an electric field between at least two electrodes [4]. From exposure to the applied electric field, biological cells will experience a rise in their transmembrane voltage to a critical value after a characteristic charging time ($\tau_m$), where pores are formed thereafter [7]. At sufficiently high intensities, these pores can fail to reseal, inducing cell death, in what is known as irreversible electroporation (IRE) [8, 9]. IRE has been used in food processing for bacterial inactivation in milk [10, 11], juices [12–14], and water [15, 16], among other media. PEF offers several advantages for food treatment over other technologies. Aside from intense exposures, PEF is a non-thermal technology, so food qualities like color, taste, texture, and nutritional composition are conserved [17]. Short processing times are required and exposures can be done in continuous flow so there is an easy implementation into existing processing lines [18]. There is also potential for the biosynthesis of vital compounds like polyphenols and carotenoids after PEF exposures [19]. There is a need for improved decontaminating technologies, as foodborne illness still affects around 600 million people and causes 420,000 deaths worldwide each year [20]. Among outbreaks involving bacteria in the United States, *Escherichia coli* (*E. coli*), *Salmonella*, and *Listeria* are three of the leading causes [21, 22].

PEF applied using pulses of nanosecond duration (nsPEF) have been recently investigated for lethal applications such as tumor ablation [23–26], electrochemotherapy [27–29], bacterial inactivation in biofilms [30] or in their planktonic state [11, 31], and the inactivation of other biological agents like viruses [32, 33]. Because of the short pulse durations, sublethal nsPEF exposures are easily attainable and several applications for them exist as well, which often involve cell stimulation or triggering of biochemical pathways for quicker or enhanced processes [19, 34].

When delivering pulses in the nanosecond range, it is important to take note of $\tau_m$, which will increase with cell radius and decrease with higher extracellular conductivity [7]. At pulse durations ($t_p$) sufficiently above the charging time, around 2–10 times longer for efficient electroporation [35], the traditional mechanism of electroporation is observed. However, at $t_p$ below $\tau_m$, intracellular contents may be disrupted instead of the outer membrane [9, 36, 37]. The short-circuiting effect of the membrane capacitance allows for penetration of the electric field to inner parts of the cell at high frequencies, typically in the MHz range [38]. In order for these effects to be seen, much higher field strengths are required than those for traditional electroporation [39, 40].

Pulses of ns duration, regardless of mechanism, will require higher field strengths for significant levels of inactivation to be observed in comparison to pulses of μs or ms duration, since longer pulse durations result in more lethal exposures, even when applied for the same total treatment times [41]. In theory, the required electric field strength to reduce an *E. coli* population by 1 log using a single monopolar 600 ns pulse is ~70 kV/cm, compared to ~ 10 kV/cm for a 100 μs pulse [7], though delivering a train of successive pulses will increase treatment efficacy [42]. For example, Martens et al. observe ~ 1 log reduction of *E. coli* using 1,000 monopolar 600 ns pulses at 13.5 kV/cm or 100 pulses at 18.5 kV/cm [31].

The correlation between increasing treatment efficacy with increasing the total number of applied pulses may be dependent on several factors. Silve et al. propose an electrodesensitization theory, in which existing, conductive pores formed in the beginning portion of a train of pulses will inhibit further charging of the membrane, thus desensitizing cells to following pulses and decreasing the efficacy of the latter portion of an exposure [43]. This has been shown with applied pulses to *Salmonella* Typhimurium, where increased treatment time correlates to a more lethal treatment, and upon a threshold value, begins to tail off, losing its efficacy

[44]. The electrosensitization theory, on the other hand, proposes that cells are sensitized to subsequent pulses [45], and more effective treatments using split-dose deliveries on mammalian cells are noted in several studies [45–48]. The efficacy of extended or delayed treatments appears to be dependent on the state of the membrane and cell post-exposure.

One distinct advantage that nanosecond pulses can offer compared to those on the micro- and millisecond timescale is lower energy input per pulse, which translates to lower costs and temperature rises [49]. Although more challenging, it is desirable in these regards to reduce the necessary power output to achieve similar effects. Another key advantage is the limiting of electrolysis, which typically occurs for water at pulse durations $> 10$ μs [50] and will protect both adverse changes in quality/composition to the treated sample and increase the lifespan of the electrodes through diminished corrosion and fouling [51]. Complicating these features are reduced efficacy of the treatments in achieving bacterial disinfection standards, commonly defined as $\geq$5-log reduction in colony forming units (equal to inactivation of 99.999% of bacteria in the sample).

Traditional PEF exposures involve a single train of pulses being delivered to a suspension. Split-dose exposures deliver half of the pulses, and then after a delay, the other half are applied. The same total energy is being delivered, but by using split-dose PEF, a delayed sensitization to the second train of pulses can be induced and can provide PEF treatments of similar or greater efficacy using lower voltages or doses [46, 48]. This can reduce operating costs, allow for greater electrode spacing, and limit ohmic heating, among other benefits. Despite the potential advantages only one study to date describes the application of split-dose PEF for the inactivation of bacteria [44], which does not investigate effects in combination with an additive. Recent innovations to improve nsPEF lethality include combination of this treatment with chemical disinfectants. Some studies include the investigation of chemical additives including antibiotics [52], bacteriocins [53], and carboxylic acids [54]. These papers note increased permeability after PEF allows for higher amounts of disinfectant to target intracellular contents, which is especially useful when diffusion across the membrane is low or nonexistent for viable cells. While selected chemical additives were found to be synergistic in bacterial inactivation, other treatments were not; suggesting that the identification of desirable synergistic effects for the combination treatment protocols are dependent on a complex combination of factors [4, 52–54].

One chemical disinfectant that has drawn interest of late as an alternative to current measures is hydrogen peroxide ($H_2O_2$), which has already been approved by the US Food and Drug Administration (FDA) for different food applications such as milk treatment in cheesemaking [55, 56]. Selected additional recent applications of $H_2O_2$ as a disinfectant in food production includes the pasteurization of dairy and eggs [57–59], decontamination of meat products [60–62], fruits/vegetables [63–66], and drinking water treatment [67–70]. Two of hydrogen peroxide's major disadvantages are that high concentrations are typically required for disinfection and low activity can be noted when in an environment with catalase-producing bacteria [71]. By combining this disinfectant with PEF, we aim to alleviate both of these issues by facilitating hydrogen peroxide's mechanism of action against microbes through pore formation, lowering the required bactericidal concentration and exposing a greater amount of $H_2O_2$ to cells before decomposition via catalase.

One ongoing challenge to employing $H_2O_2$ as disinfectant is the observation of undesired impacts on food products at concentrations suitable for disinfection resulting in fruit and vegetable browning, bleaching, and other negative effects. Catalase enzyme washes and other methods can be employed to limit these effects [72–74]. Current processes employing hydrogen peroxide as an antimicrobial agent vary in their maximal allowed treatment concentrations. For example, milk may be treated with up to 0.05% and starch up to 0.15% (21CFR184.1366) in accordance with the FDA. In almost all cases, hydrogen peroxide must be

removed using proper chemical or physical methods. Residual H$_2$O$_2$ in food packaging may not exceed 0.5 ppm (21CFR178) in accordance with the FDA.

In addition to direct food applications, H$_2$O$_2$ could serve as a biocide for irrigation water [75]. Multiple outbreaks in recent years have occurred where irrigation water was identified as a potential source [76, 77]. Waste and surface water are economically feasible sources for irrigation that will often need to be treated in accordance with the new Codex standard (XG 100–2023) to reduce contamination risks. H$_2$O$_2$ is considered to be generally recognized as safe (GRAS) because of its eventual degradation into water and oxygen [78] so its environmental impact is limited. It offers many advantages as a sanitizing agent, including lack of toxicity, limited changes to food properties, and no color or odor, among others [71]. In current processes employing hydrogen peroxide as a chemical disinfectant, one of the most common concentrations used for microbial inactivation is 3% (w/v) [79, 80]. H$_2$O$_2$ is used in nonfood-related applications as well. For example, H$_2$O$_2$ has been investigated for its use in decontaminating therapeutic pool waters to decrease the risk to bathers' health [81].

We hypothesize that by employing PEF in combination with H$_2$O$_2$, it will lower the required amount of oxidant needed and will increase the effectiveness of the treatment for inactivation of bacteria.

Herein, we describe our investigations of a widely applicable methodology for the combination of chemical disinfection using hydrogen peroxide with nsPEF. We describe an optimized combination treatment effective for the inactivation of several bacteria species. This study further demonstrates the synergistic effects between split-dose and traditional nsPEF exposures in combination with sub-inhibitory (less than 0.5% w/v) concentrations of hydrogen peroxide. Accordingly, our investigations provide guidance for the inactivation of bacteria in Gomori buffer with the administration of exceptionally low concentrations of H$_2$O$_2$.

## Results

### PEF treatment profile

We evaluated nsPEF sequences up to 3000P for inactivation of *E. coli*, *L. innocua*, and *S. enterica*. In this experiment, we observed dose dependent increase in bacterial disinfection correlated to number of pulses approaching a plateau of maximal reduction in bacterial CFU at ~1000P (S1 Fig). The plateau of maximal efficacy of nsPEF is genera dependent. The greatest effect for this treatment is observed for *E. coli*, resulting in 0.87±0.08 log CFU reduction for 1000P. The nsPEF treatment was less effective for *L. innocua* and *S. enterica*, providing log CFU reductions of 0.13±0.06 and 0.25±0.04 (P<0.0001) for these genera, respectively. This is in agreeance with the commonly noted 'tailing' effect in inactivation curves when increasing the number of applied pulses [44, 82]. Accordingly, we focused our attention on the identification of a combination treatment for bacterial disinfection employing a PEF pulse sequence of 1000 total pulses.

### H$_2$O$_2$ disinfectant profile

We sought to establish the dose-dependent reduction of bacterial CFU under the experimental conditions employed in this study for H$_2$O$_2$. The different bacterial species evaluated show different dose-dependent effects. We find that H$_2$O$_2$ shows the greatest potency against *E. coli* and causes greater changes in viability with lower concentrations and contact times. *S. enterica* and *L. innocua* are more resistant to H$_2$O$_2$ and show little to no reduction in viability from our employed concentrations (0.5%, 0.3%, and 0.1%). At concentrations $\geq$ 1%, viability changes drastically. EC$_{50}$ values were determined to be 0.96% H$_2$O$_2$ for *E. coli* after 15 min, 0.58% H$_2$O$_2$ for *S. enterica* after 45 min, and 0.87% H$_2$O$_2$ for *L. innocua* after 45 min. For the tested contact times, a 5-log reduction is reached after incubation in 2% H$_2$O$_2$ for *E. coli* and *L.*

*innocua* and in 1% H$_2$O$_2$ for *S*. enterica (S2 Fig). For combination studies with PEF, we selected three H$_2$O$_2$ concentrations below the EC$_{50}$ value for treatment with H$_2$O$_2$ alone.

## PEF combined with H$_2$O$_2$

Three variables are examined to improve inactivation efficiency of our combination treatment: contact time, H$_2$O$_2$ concentration, and PEF exposure type. These variables were systematically evaluated for the inactivation of *E. coli*, *L. innocua*, and *S. enterica* (Table 1). For each treatment condition triplicate analysis was performed with good reproducibility. Greater variability is noted in selected examples in which the treatment conditions were less effective, under maximally effective treatment conditions no bacterial growth was observed in any of the replicates after 16-32h incubation on LB agar plates. We observe statistically significant impact from both changes in PEF and incubation time at each concentration of H$_2$O$_2$ evaluated. Graphical representation of this data, including representation of statistical differences from two-way ANOVA and Tukey's multiple comparisons test, are included in Fig 1. The full tabulated statistical analysis is included in the Supplemental (S2 Table).

Evaluation of disinfection for each bacterial species with H$_2$O$_2$ or PEF alone compared to the combination treatment demonstrates the significant benefit for the combination treatment (Fig 2). Indeed, for each of the bacterial species evaluated the combination of PEF and low concentrations of H$_2$O$_2$ provided synergistic levels of bacterial inactivation.

The effectiveness of PEF treatments varies between the different bacteria, however it is impacted in a dose-dependent manner correlated to contact time for all bacterial species evaluated. *E. coli* reaches a 5 log reduction after 15 min at 0.1% H$_2$O$_2$ with only the split-dose exposure but does so for both exposure types after the same contact time at higher concentrations (0.3% and 0.5% H$_2$O$_2$). Split-dose exposures show significant increases of inactivation compared to single trains after 15 min at 0.1% H$_2$O$_2$ and 10 min at 0.3% H$_2$O$_2$ but displays similar effectiveness at 0.5% H$_2$O$_2$ for all examined contact times. Of note, *E. coli* is the most susceptible to hydrogen peroxide of the three bacteria in this study in the absence of PEF. While less than 1 log reduction inactivation is found for *L. innocua* and *S. enterica* at any concentration of H$_2$O$_2$ without PEF treatment, *E. coli* is reduced by 0.8 and 2.5 log at 0.3% and 0.5% H$_2$O$_2$, respectively, after 15 min.

Unlike with *E. coli*, a 5 log reduction is not achieved at 0.1% H$_2$O$_2$ for either of the other bacteria evaluated in this study. We find that *L. innocua* and *S. enterica* both require minimally 0.3% H$_2$O$_2$ and longer contact times to reach this level of disinfection. *L. innocua* and *S. enterica* do however follow a similar trend to *E. coli* with enhanced effect of split-dose PEF for contact times below 30 minutes (Fig 2). At these shorter contact times for these two bacteria, less than 5 log reduction in CFU is observed. Additionally, for these contact times we observe enhanced disinfection in the samples treated with the split-dose PEF treatments by comparison to the single-dose PEF treatment. By contrast, with longer contact times the additional effect of the split-dose PEF becomes less prominent and similar levels of log CFU reduction are observed for contact times of 30 and 45 minutes with the single-dose and split-dose PEF. Interestingly, for *S. enterica*, certain treatments show the opposite effect to the trend of enhanced reduction of bacterial load with the split-dose PEF. Specifically, at 0.3% H$_2$O$_2$ and a contact time of 45 min the split dose PEF treatment achieves only 3.4 log reduction in CFU whereas the single dose PEF provides 5 log reduction. Similarly, at a concentration of 0.5% H$_2$O$_2$ and a contact time of 30 min, split-dose PEF results in 4.6 log reduction while the single dose PEF gives 6 log reduction.

To evaluate effects on cell structure, bulk effects on cell shape or aggregation during the treatments, selected samples were evaluated by transmission light microscopy. No cell

**Table 1. Bacterial inactivation of *E. coli*, *L. innocua*, and *S. enterica* based on CFU determination.**

| % H$_2$O$_2$ | Contact Time | 0P[a] | 1000P[b] | 500+500P[c] |
|---|---|---|---|---|
| | | *Escherichia coli* | | |
| 0.1 | 5 min | 0.01±0.06 | 2.40±0.20 | 2.56±0.16 |
| 0.1 | 10 min | 0.09±0.16 | 2.88±0.05 | 3.28±0.58 |
| 0.1 | 15 min | 0.22±0.07 | 3.69±0.86 | **5.80±0.01** |
| 0.3 | 5 min | 0.09±0.08 | 2.03±0.12 | 2.09±0.13 |
| 0.3 | 10 min | 0.24±0.02 | 2.56±0.59 | 4.73±1.24 |
| 0.3 | 15 min | 0.76±0.09 | **5.52±0.49** | **5.64±0.28** |
| 0.5 | 5 min | 0.23±0.01 | 1.37±0.09 | 1.72±0.18 |
| 0.5 | 10 min | 1.24±0.07 | 3.75±0.97 | 4.31±0.44 |
| 0.5 | 15 min | 2.48±0.47 | **5.80±0.00**[d] | **5.80±0.00**[d] |
| | | *Listeria innocua* | | |
| 0.1 | 5 min | 0.16±0.07 | 0.22±0.07 | 0.27±0.10 |
| 0.1 | 10 min | 0.09±0.05 | 0.22±0.02 | 0.59±0.11 |
| 0.1 | 15 min | 0.18±0.04 | 0.29±0.07 | 0.70±0.07 |
| 0.1 | 30 min | 0.02±0.02 | 0.68±0.13 | 1.50±0.15 |
| 0.1 | 45 min | 0.01±0.04 | 0.76±0.06 | 1.59±0.06 |
| 0.3 | 5 min | 0.13±0.05 | 0.29±0.02 | 0.40±0.05 |
| 0.3 | 10 min | 0.10±0.03 | 0.49±0.13 | 1.51±0.46 |
| 0.3 | 15 min | 0.14±0.03 | 0.60±0.05 | 1.77±0.18 |
| 0.3 | 30 min | 0.02±0.02 | 2.87±0.55 | 2.98±0.31 |
| 0.3 | 45 min | 0.08±0.02 | **5.70±0.47** | **5.56±0.81** |
| 0.5 | 5 min | 0.09±0.05 | 0.31±0.12 | 0.39±0.07 |
| 0.5 | 10 min | 0.02±0.02 | 0.40±0.27 | 1.99±0.18 |
| 0.5 | 15 min | 0.03±0.04 | 1.18±0.32 | 2.75±0.17 |
| 0.5 | 30 min | 0.08±0.06 | **6.24±0.00**[d] | **5.91±0.58** |
| 0.5 | 45 min | 0.12±0.04 | **6.24±0.00**[d] | **6.24±0.00**[d] |
| | | *Salmonella enterica* | | |
| 0.1 | 5 min | 0.04±0.01 | 0.41±0.03 | 0.36±0.06 |
| 0.1 | 10 min | 0.02±0.01 | 0.37±0.11 | 0.77±0.05 |
| 0.1 | 15 min | 0.03±0.02 | 0.49±0.07 | 1.19±0.57 |
| 0.1 | 30 min | 0.03±0.06 | 0.49±0.16 | 1.93±0.36 |
| 0.1 | 45 min | 0.00±0.06 | 0.75±0.04 | 2.27±0.13 |
| 0.3 | 5 min | 0.03±0.01 | 0.17±0.02 | 0.36±0.08 |
| 0.3 | 10 min | 0.04±0.06 | 0.52±0.27 | 1.66±0.45 |
| 0.3 | 15 min | 0.05±0.03 | 0.96±0.26 | 2.54±0.11 |
| 0.3 | 30 min | 0.03±0.04 | 2.27±0.14 | 2.80±0.41 |
| 0.3 | 45 min | 0.05±0.03 | 4.96±0.15 | 3.40±0.19 |
| 0.5 | 5 min | 0.01±0.02 | 0.45±0.06 | 0.44±0.08 |
| 0.5 | 10 min | 0.03±0.02 | 1.91±1.12 | 1.90±0.49 |
| 0.5 | 15 min | 0.02±0.04 | 3.05±0.66 | 2.65±0.16 |
| 0.5 | 30 min | 0.10±0.04 | **5.95±0.17** | 4.60±0.16 |
| 0.5 | 45 min | 0.89±0.02 | **5.72±0.58** | **5.89±0.27** |

All data is reported as average log reduction in bacterial CFU of triplicate analysis with error representing standard deviation. Conditions resulting in > 5-log reduction in CFU are shown in bold.

[a] No PEF performed.

[b] Single train PEF treatment.

[c] Split-dose PEF treatment.

[d]No bacterial colonies observed in any of the replicates during enumeration following treatment.

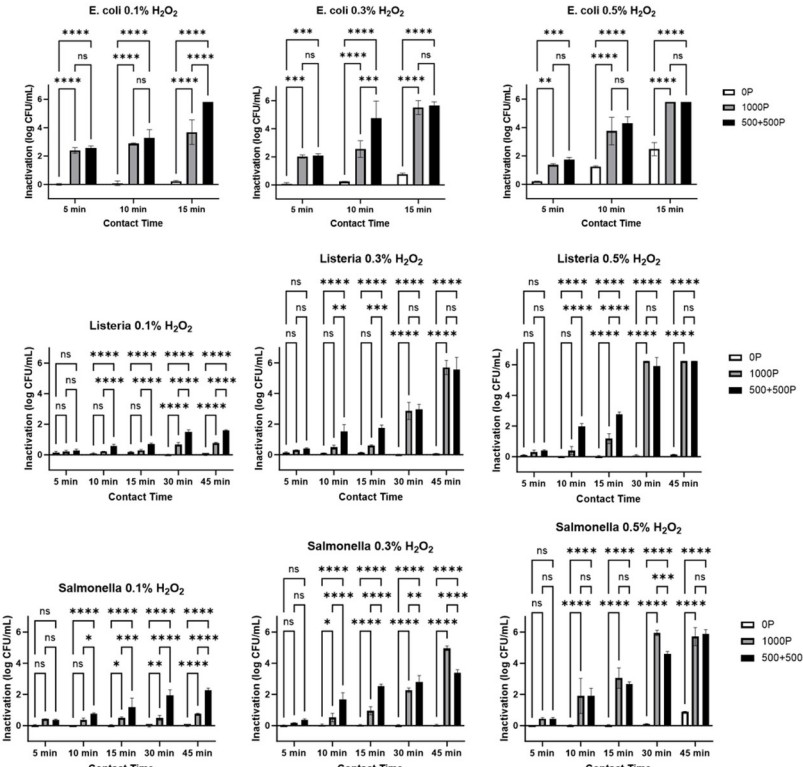

**Fig 1. Log CFU reduction of *E. coli*, *L. innocua*, and *S. enterica* varying pulsing conditions, contact times, and concentrations of hydrogen peroxide.** Data presented as average of triplicate analysis with error bars representing standard deviation. Statistical differences from Tukey's multiple comparisons test are presented as (ns) P > 0.05, * P < 0.05, ** P < 0.01, *** P < 0.001, **** P < 0.0001.

clumping or obvious morphological changes to the cell structures were observed under conditions leading to partial disinfection (S3 Fig).

## Electrosensitization

For *E. coli* and *L. innocua* we observe enhanced disinfection for treatments employing the split-dose PEF (500+500 pulses) compared to the single-dose PEF (1000 pulses) at all concentrations of H$_2$O$_2$ (Fig 1). By contrast, disinfection of *S. enterica* cultures were not universally enhanced by split-dose PEF. To evaluate the effect of the single-dose compared to the split-dose PEF treatments we evaluated the impact of PEF delay time on bacterial viability in the absence of H$_2$O$_2$ (Fig 3). As anticipated, *E. coli* susceptibility is correlated with increased delays between PEF trains, with a maximum inactivation increase of around 40% after a 30 min delay. *S. enterica* and *L. innocua* show much greater resistance to PEF treatments in general. Despite this, *L. innocua* appears to display higher sensitivity to the exposures with an increasing delay reaching a maximal sensitivity at 27 min. *S. enterica* does not appear to display any relationship between lethality enhancement and exposure delay.

## Discussion

Hydrogen peroxide can function as an effective bactericidal agent because of its strong oxidating properties. The formation of short-lived hydroxyl radicals occurs most commonly because

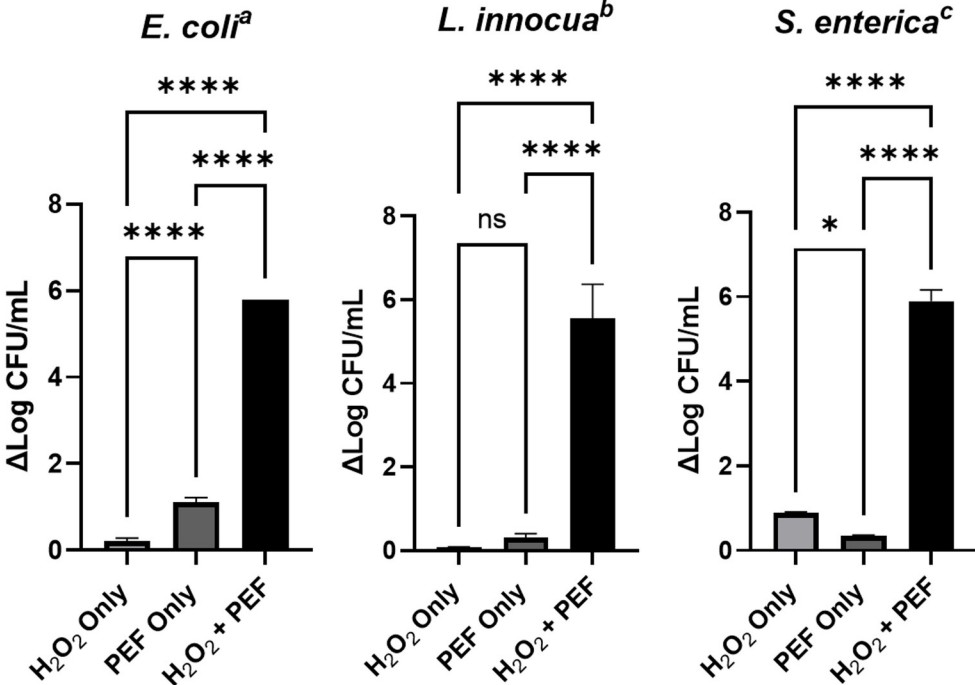

**Fig 2. Comparison of inactivation using individual treatments and the combination H$_2$O$_2$/PEF treatment.** All data represents the average of triplicate analysis, error bars represent standard deviation. Statistical significance from Tukey's multiple comparisons test; (ns) P > 0.05, * P < 0.05, ** P < 0.01, *** P < 0.001, **** P < 0.0001 [a] *E. coli* treatment conditions: 0.1% H$_2$O$_2$, 15 min contact time, split-dose PEF (500+500P). [b] *L. innocua* treatment conditions: 0.3% H$_2$O$_2$, 45 min contact time, split-dose PEF (500+500P). [c] *S. enterica* treatment conditions: 0.5% H$_2$O$_2$, 45 min contact time, split-dose PEF (500+500P).

of the Fenton reaction [83, 84], which involves the oxidation of Fe$^{2+}$ by H$_2$O$_2$. Although particularly reactive with iron, hydrogen peroxide can be reduced by other transition metals as well, such as copper [85]. Electrode material can therefore play an important role in catalyzing this reaction and be susceptible to degradation. The use of aluminum electrodes in this study, however, has a negligible effect on H$_2$O$_2$. Aluminum is one of the most compatible materials with H$_2$O$_2$, even at high concentrations [86], because of an aluminum oxide layer that forms and prevents corrosion of the metal [87].

With a compatible electrode material and a buffer consisting of no added transition metal ions, the H$_2$O$_2$ reduction must be catalyzed by bioavailable ions. It is estimated that a third of all proteins contain metal [88] and 40% of enzymes with known structures are metal-dependent [89]. While most of the total metal content within a bacterium cell is bounded, free metal ions also exist on the μM scale [90]. H$_2$O$_2$ will specifically oxidize enzymes and proteins complexed to iron in addition to Fe$^{2+}$ ions within the labile pool [91]. Hydroxyl radicals have been shown to be effective for microbial decontamination in food processing settings [92, 93]. Pore formation from PEF will cause either the efflux of intracellular contents or the influx of H$_2$O$_2$, exposing the chemical to a much higher concentration of metal ions compared with diffusion. The quicker formation of hydroxyl radicals will generate more significant damage to the surrounding proteins, enzymes, lipids, and DNA [84, 85, 91, 94]. Based on this, hydrogen peroxide would ideally be added to a production line soon before PEF exposures. This maximizes H$_2$O$_2$ uptake into cells through pores and minimizes the amount molecules that are degraded by catalase.

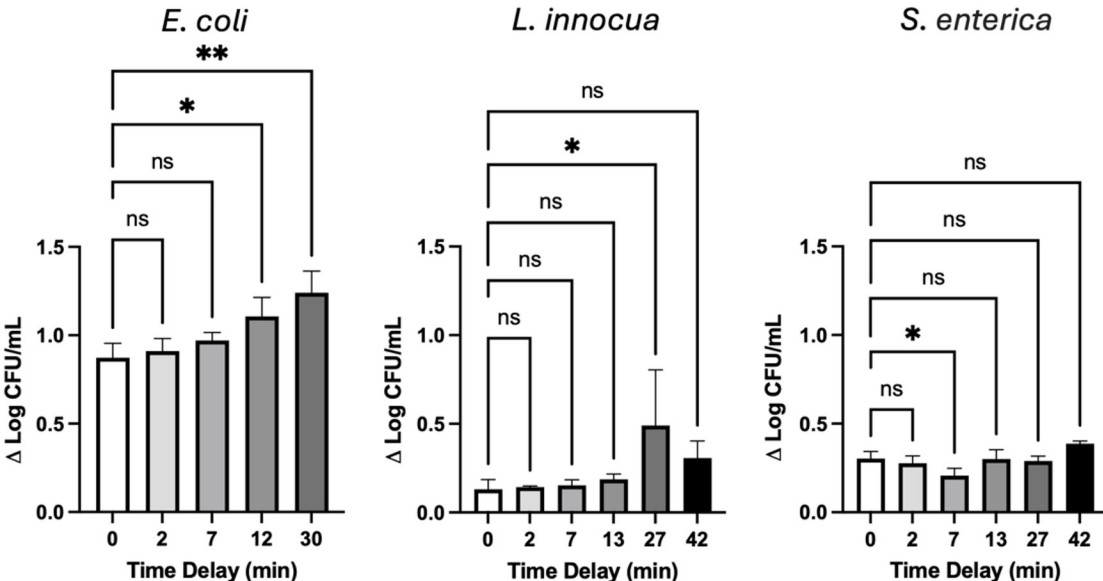

**Fig 3. Change in log reductions of different bacteria after varying time delays between two trains of 500, 600 ns pulses at 21 kV/cm in the absence of H$_2$O$_2$.** The delay times between trains of PEF were selected to correlate with the contact times evaluated for combination treatments. A delay time of 0 represents a single train of 1000 pulses. Data is presented as the average of triplicate analysis with error bars representing standard deviation. Statistical differences from one-way ANOVA are presented as (ns) $P > 0.05$, * $P < 0.05$, ** $P < 0.01$, *** $P < 0.001$, **** $P < 0.0001$.

The observed results strongly support the application of split-dose PEF for bacterial inactivation. Previous work [44] had examined the application of split-dose micro-second pulses for the inactivation of *S. Typhimurium*. Here, we expand on this finding demonstrating a strategy for the enhancement of PEF efficacy without increasing the applied energy by employing split-dose nanosecond pulses in combination with H$_2$O$_2$.

In our study, split-dose treatments increase PEF efficiency in both Gram-positive and Gram-negative strains. The observed broad spectrum antimicrobial effects suggest a general mechanism of action for the treatment, however, the electrosensitization phenomenon alone does not explain the higher reductions seen in many instances when combined with H$_2$O$_2$. Lipid peroxidation is a result of reactive oxygen species (ROS), e.g., hydroxyl radicals, interacting with biomembranes, and is associated with increased membrane fluidity [95–97]. PEF exposures are often more effective at higher temperatures where membrane fluidity is also increased [98–100]. Although it is proposed that alterations in PEF resistance is more complex than simply examining changes in fluidity [98, 99], it may be a contributing factor. Future studies should incorporate scanning electron microscopy to assess membrane alterations. In a general sense, we hypothesize that oxidative damage to the membrane structure induces greater susceptibility to PEF [101].

There may be a relationship, however, between contact time and hydrogen peroxide concentration for the effectiveness of split-dose nsPEF treatments (Fig 4). Efficacy of these exposures vary, appearing to tail with time in some instances and insufficiently boost inactivation compared to their single train counterparts in others. Based on this study, some patterns can be identified that appear to be species independent. For combination treatments involving 0.1% H$_2$O$_2$ (Fig 4, blue) increasing benefit for split-dose treatments is observed with increasing contact times. For treatments with 0.3% H$_2$O$_2$ (Fig 4, green) we observe initially increasing benefit for split-dose nsPEF with increasing contact times. However at longer contact times

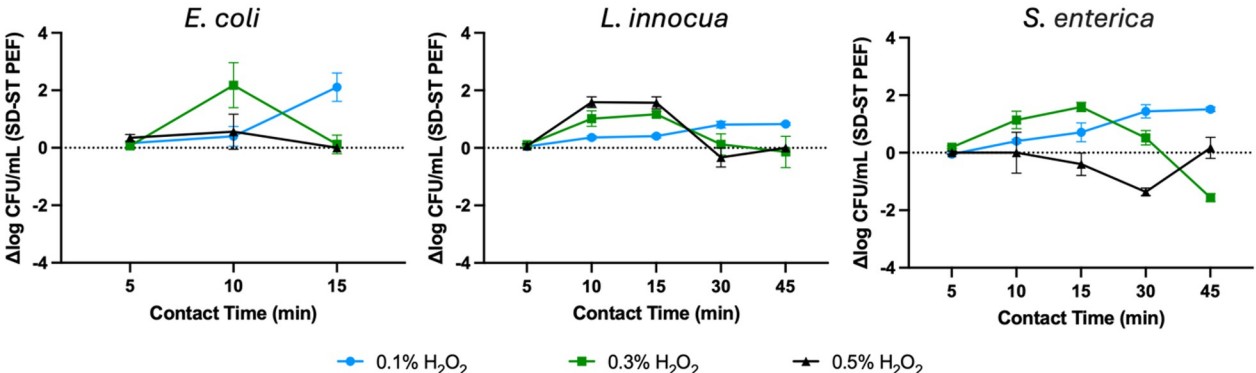

**Fig 4. Difference in log CFU reductions in bacterial load between split-dose (SD) and single train (ST) exposures for bacteria.** Positive values represent enhanced log CFU reduction for split-dose PEF compared to single train exposures, negative values represent enhanced log CFU reduction for single train PEF exposures compared to split-dose exposures.

this effect returns to zero, or in the case of *S. enterica* shifts to a benefit for the single-train nsPEF at the longest contact time. For treatments with 0.5% H$_2$O$_2$ (Fig 4, black), a pattern is less apparent. While *E. coli* treatments show equally effective disinfection with both single-train and split-dose nsPEF at 0.5% H$_2$O$_2$, *L. innocua* displays a polymodal profile similar to this bacteria with 0.3% H$_2$O$_2$ and *S. enterica* appears to display an inversion of this polymodal profile, initially showing enhanced potency of single-train before returning to no difference between nsPEF treatments.

While evidence suggests oxidative damage aids in enhancing susceptibility to PEF exposures, longer H$_2$O$_2$ exposures may be inducing morphological effects that actuate the contrary. Significant reductions in volume have been shown for *E. coli* at H$_2$O$_2$ concentrations > 10 mM [102]. It is well known that smaller cells have less susceptibility to PEF due to decreased exposure areas [103]. Higher concentrations and longer contact times may sufficiently shrink cells to raise the electroporation threshold and increase resistance to treatments. Reduction in cell volume may also improve cellular ability to regenerate outer layer damages [104]. Oxidative damage is also linked to membrane depolarization [105, 106]. During an electric pulse, cell membranes will be hyperpolarized at the side facing the anode and depolarized at the side facing the cathode [107], where under hyperpolarizing conditions, larger pore size [108] and density [109] are noted. With nsPEF exposures already lacking production of large pore formation [110], inhibition or delay of hyperpolarization on the anodic facing portion of the membrane could further prompt a population of smaller pores or less pores all together, increasing likelihood for survival. Clumping of microbes may also reduce the effectiveness of PEF exposures [82] although that does not appear to be an influence in this study based to our observations with light microscopy (S3 Fig).

In general, Gram-positive bacteria show greater resistance to PEF exposures than Gram-negative [111–114]. Our observations are consistent with this when comparing *E. coli* and *L. innocua*, so a lesser degree of H$_2$O$_2$ uptake may be part of the reason longer contact times are required to achieve sufficient log reductions in *L. innocua*. The thicker peptidoglycan layer of the Gram-positive *L. innocua* may also quench hydroxyl radical production [115]. *Listeria* has also been shown to have greater catalase activity than *E. coli* [116], so a greater chance exists that hydrogen peroxide is quenched by this enzyme, thereby preventing hydroxyl radical formation. *S. enterica* may be showing greater PEF resistance due to a greater membrane charging time constant. Although, as previously stated, larger cells are more susceptible to PEF

exposures, in a low conductivity buffer, and with pulses of ns duration, charging time becomes a more prominent factor in determining extent of electroporation. The calculated $\tau_m$ for *E. coli* is 66 ns, about 9.1x less than the pulse duration used in this study. For *S. enterica* the calculated $\tau_m$ is more than double *E. coli*'s, and only about 4.4x the pulse duration. 95% membrane charging capacity is not achieved until ~ $3\tau_m$ [34], so it is anticipated that *E. coli* is charged at a higher capacity for longer than *S. enterica* possibly explaining the diminished PEF efficiency against the latter. *L. innocua* and *S. enterica* display similar resistances to PEF in this study, as well as comparable reductions in populations amongst different concentrations and contact times. Taken together, our data supports the hypothesis that the extent of pore formation is a major consideration for the efficacy of combination treatments of hydrogen peroxide and nsPEF.

While higher bacteria innoculation would not be expected to result in less effective PEF exposures [117], it could affect the activity of H$_2$O$_2$. *L. innocua* and *S. enterica*, were inoculated at higher initial bacteria populations compared to *E. coli* ($10^8$ vs. $10^7$ CFU initial inoculations), so H$_2$O$_2$ availability to cells may decrease and total catalase in solution could increase. This could also be a factor, in addition to PEF efficiency, for why longer contact times and higher concentrations of hydrogen peroxide are required for effective inactivation to be noted for *L. innocua* and *S. enterica* compared to *E. coli*.

Pore resealing occurs in three stages: rapid reduction in size (μs–ms), slower reduction to a size of about 0.5 nm (min), and complete closure (min–h) [118]. At higher frequencies, pulses may be delivered too quickly for proper resealing and for sensitization to develop. Enhanced lethality is observed with treatment delays of a few minutes for mammalian cells, going as low as 30 s [48]. The required delay will depend on post-exposure conditions, such as temperature and suspension media, which will dictate resealing kinetics. Factors for why sensitization occurs can include energy expenditure after pore resealing, altered cell physiology, oxidative damage, cytoskeleton rearrangement, or swelling [45, 101, 119, 120]. Delso et al. attribute delayed sensitization primarily to the cellular ability to reseal pores in varying conditions for Gram-negative *S. enterica* [44]. The rigidity and stability of the outer peptidoglycan layer of Gram-positive bacteria may inhibit structural changes contributing to sensitization of Gram-negative bacteria. A lesser extent of pore formation may also prevent osmotic swelling and prevent volumetric changes [121]. Electrosensitization likely occurs due to a multitude of factors, and the mitigation of some could be reasoning for a longer required delay.

In this study, we find that both *S. enterica* and *L. innocua* exhibit reduced electrosensitization compared to *E. coli*. PEF resistance is greater amongst these two strains, indicating that the first train needs to be sufficiently intense for the second to benefit. Delaying PEF treatment appears to not have a considerable effect in heightening the effectiveness of unfavorable pulsing parameters (in terms of lethality). Beyond studies of the rate of pore formation, pore resealing kinetics and their role in electrosensitization for Gram-positive and Gram-negative bacteria, represents an important area for future investigations, especially using sub-μs pulses at higher field strengths.

While increased temperature can greatly impact bacterial survival on its own, it can also enhance the biocidal activity of H$_2$O$_2$ by reducing necessary exposure times [122]. The role of Joule heating in this study, however, appears to be negligible based on COMSOL modeling with experimental validation. The PEF conditions employed in this study are observed to elevate temperature inside the cuvette by <2°C from ambient temperature (~23°C). Given that these bacteria are cultured at 37°C, the observed temperature increase from PEF is insignificant with respect to bacterial viability in this study. Not surprisingly, we find that PEF alone is ineffective at bacterial inactivation (Fig 2).

Other studies have combined technologies with hydrogen peroxide addition. UV light was shown to be ineffective within the range of H$_2$O$_2$ concentrations tested (0.003–0.015%) [123],

cold atmospheric pressure plasma jet did not note a 5 log reduction with higher H$_2$O$_2$ concentrations of < 1.0% H$_2$O$_2$ [124], and ultrasound was unable to attain a 5 log reduction in combination with 0.3% H$_2$O$_2$ [125]. The conductivity of the buffer employed in our studies is comparable to tap water, further enhancing the potential application of this technology in the food industry, especially for bacterial decontamination of plant cells with harder shells that can withstand high intensity, monopolar treatments, like seeds. PEF has been shown to be effective for this treatment on several seed types, including lettuce and arugula [126]. The use of this technology may also serve as a viable option in wastewater treatment [127, 128]. Using H$_2$O$_2$ would reduce unwanted byproducts compared to other water treatment techniques like chlorination [16]. Leveraging the disinfection strength of the described split-dose nanosecond PEF/ H$_2$O$_2$ treatment conditions may also enable control of resistant cell types including bacterial spores [129, 130]. PEF treatment alone has been shown to be ineffective against bacterial spores [131] however ≥1% H$_2$O$_2$ in combination with other treatments has been observed to be sporicidal [132, 133]. Future studies exploring the sporicidal activity of the H$_2$O$_2$/PEF treatment described in this manuscript and potential applications in wastewater treatment are warranted [16, 81].

In summary we demonstrate a combination hydrogen peroxide/nsPEF methodology for the treatment of bacterial contamination. We demonstrate strong synergy between the combination treatments resulting in highly effective decontamination of multiple bacterial species (S7 Fig). The optimized methodology requires the addition of only low (down to 0.1% w/v) concentrations of the GRAS additive hydrogen peroxide to provide up to 5 log reduction in bacterial CFU. We hypothesize that oxidative damage from the addition of low concentrations of H$_2$O$_2$ increases bacterial susceptibility to PEF exposures. However, we note that this effect does appear to elicit a tolerance that is dependent on both contact time and H$_2$O$_2$ concentration. We, for the first time, demonstrate that split-dose nanosecond exposures enhance efficacy of the PEF treatment in both gram-positive and gram-negative bacteria. In this study we further characterize the significance of delay time between pulse trains. The biomolecular cause for the species-specific impact of delay time between PEF represents an important area for future investigations. We observe that increasing the delay time between nsPEF results in greater reduction in CFU for *L. innocua* and *E. coli* however this effect was not observed for *S. enterica*. Combination of nsPEF and low concentrations of hydrogen peroxide appears to be a promising procedure for decontamination, enabling lower energy exposures of PEF and significantly decreased hydrogen peroxide concentrations.

Future investigations for applications of this technology for food products will most likely have higher conductivities than the Gomori buffer as we employ in our studies. Changes to the conductivity of the matrix will require reoptimization of treatment conditions as changes in matrix conductivity is expected to impact the efficacy of PEF. Applications in systems with conductivities greater than Gomori buffer should lower the outer membrane charging times and thereby increase the efficacy of nsPEF treatments. The critical transmembrane potential for pore formation to occur is not reached until after a characteristic charging time constant [134]. After the membrane is charged, structures destabilize and pores form and expand [135]. This can greatly alter exposure effects in the ns regime, which often operate using pulse durations near the membrane charging time. As external conductivity increases, membrane charging time decreases [8], and so ns pulses are more likely to operate sufficiently above the charging time that can induce efficient electroporation (~2–10 times higher [35]). We note that increased conductivity would result in increased current/energy inputs and greater temperature increases with PEF and therefore may require additional engineering considerations for application [51]. The quality of food products may also be impacted by hydrogen peroxide, although the demonstrated use of very low H$_2$O$_2$ concentrations is anticipated to mitigate this

effect. Finally, for any specific application it is anticipated that a broad range of potential methods for bacterial inactivation will need to be evaluated. Accordingly, the described synergistic combination of H$_2$O$_2$ and PEF described in this manuscript thereby provides an additional potent technology for the control of bacteria.

It is important to consider the cost of this technology in potential future industrial applications. Despite hydrogen peroxide's many advantages, it can be more expensive compared to other disinfectants [136]. Based on the results of this study, necessary concentrations can be greatly reduced for microbial inactivation, which would lower the financial limitations of H$_2$O$_2$ use. A mixture of H$_2$O$_2$ and peracetic acid (PAA) forms a strong oxidizer shown to be an effective decontaminant in several food types [137] that is stronger than either alone and compared to many other commercial sanitizers [56]. The synergy between the two chemicals, in addition to the synergy in combination with PEF, could enhance lethality of treatments with even lower concentrations and PEF energy inputs, further lowering costs.

Based on the positive results of this study, future investigations to further characterize the role of the kinetics of both pore formation and pore resealing in nsPEF and the impact of these phenomena on the synergy between H$_2$O$_2$ observed in our work would be especially valuable. Membrane damage should be assessed for a better understanding of the mechanism. Although currently no known mechanisms for bacterial resistance to PEF treatment exist, this is an underexplored area of study [138, 139]. Finally, here we focus on the decontamination of planktonic bacteria and further advances in the application of nsPEF as a treatment for bacterial pathogens will benefit from more investigations into the potential of this method for the decontamination of bacterial biofilms [140].

## Materials and methods

### Bacteria and cultivation conditions

*E. coli* D31 [141], *Listeria innocua* (*L. innocua*) (ATCC 51742), and *Salmonella enterica* subsp. *enterica* serovar Typhimurium (*S. enterica*) (ATCC 14028) were maintained on Luria Bertani (LB) agar, Miller (BD Difco™, Sparks, MD, USA) petri dishes at 4˚C. Long-term storage of bacteria was done in a cryogenic -80˚C freezer. A single colony from the plate was inoculated in 2 mL of LB Broth, Miller (BD Difco™, Sparks, MD, USA) in a 5 mL culture tube and incubated overnight at 37 ˚C with constant agitation using a vortex mixer (Crystal Industries, McKinney, TX) until stationary growth phase was reached. All media coming into contact with bacteria was sterilized using a Tuttnauer 3850 EL autoclave (Hauppauge, NY, USA).

### Hydrogen peroxide treatments

30% (w/v) H$_2$O$_2$ (Millipore Sigma, Burlington, MA, USA) was diluted with 2 mM Gomori (1.07 mM potassium phosphate dibasic and 0.93 mM potassium phosphate dibasic) buffer to make 5% (1,633 mM), 3% (980 mM), and 1% (327 mM) H$_2$O$_2$ solutions. The Gomori buffer was pH adjusted to 7.4 using KCl and KOH and emulated a conductivity similar to tap water at 480 μS/cm. Gomori buffer was chosen to eliminate any underlying effects from the media. Phosphate buffers are common in biological applications in protecting cells from osmotic imbalances. A pH of 7.4 mimics the pH of the body and is ideal for many bacteria.

*E. coli* and *S. enterica* cultures were harvested by centrifugation at 1,400 g for 5 min and *L. innocua* at 6,000 g for 5 min. Cells were washed three times before a final resuspension in the Gomori buffer, which was then diluted to an optical density (OD) of 0.10 at 600 nm. This equated to 6.3x10$^7$ CFU/mL for *E. coli*, 1.8x10$^8$ CFU/mL for *L. innocua*, and 1.1x10$^8$ CFU/mL for *S. enterica*. 20 μL of the aforementioned H$_2$O$_2$ solutions and 180 μL of the cell suspension were mixed for final H$_2$O$_2$ concentrations of 0.1, 0.3, and 0.5%, which are ~ten-fold reductions

of common concentrations of H$_2$O$_2$ used in industrial processes. Varying contact times were investigated for the different bacteria, ranging from 5–45 min to achieve a 5 log reduction in bacterial load. All exposures were performed at ambient temperature (~23 °C).

To examine bacteria sensitivity to H$_2$O$_2$ alone, cells were also exposed to 1, 2, and 3% H$_2$O$_2$ solutions at the highest contact times tested for PEF exposures (15 minutes for *E. coli*, 45 min for *L. innocua* and *S. enterica*). CFU determination was performed thereafter.

## PEF exposures

Electric pulses were delivered using a custom electroporator (S4 Fig). Exposure parameters were inputted using a BNC Model 525 Pulse/Delay Generator (San Rafael, CA, USA). This device triggered the internal MOSFET of a Behlke solid-state switch (HTS 81-06-GSM) (Billerica, MA, USA). The Extech 382200 DC Power Supply (Nashua, NH, USA) was used to power the switch, and high-voltage was fed with a GPE-3323 DC Power Supply (GW Instek, Montclair, CA, USA) in series with a potential transformer. Pulses were monitored using the DSOX1102G 70 MHz Oscilloscope (Keysight, Santa Rosa, CA, USA) and the CT3681 70 MHz Differential Probe (Cal Test Electronics, Yorba Linda, CA, USA), and ultimately delivered to a safety dome (BTX, Holliston, MA, USA) capable of harboring standard electroporation cuvettes.

Electric field strength distribution during a pulse and the temperature rise after 500 pulses was determined by COMSOL simulation (S5 and S6 Figs). Temperature change during PEF follows a linear increase after about 2 s of treatment, indicating that temperature would increase by ~1.9 °C for 1,000 P or ~5.2 °C for 3,000 P.

Experimental temperature measurements were collected for test samples of Gomori buffer to validate the predictions of the COMSOL model. Temperature readings were obtained using an Omega OM-EL-USB-TC Data Logger equipped with a Hypodermic Needle Probe (HYP1-30-1/2-T-G-60-SMP-M). Data was analyzed using EasyLogUSB Version 7.5.0.0 software and exported to Microsoft Excel. Temperature was recorded immediately before and after PEF exposures (1,000 P). The average temperature increase for six replicates was observed to be +1.8±0.5°C (S1 Table).

## Electrical exposures with additive

Cells were harvested in the same manner as the H$_2$O$_2$ treatments.

The characteristic charging times of the bacteria in Gomori buffer were found using the following equation from Schoenbach et al. [8]:

$$\tau_m = \left(\frac{\rho_s}{2} + \rho_c\right) C_m R \tag{1}$$

where $\rho_s$ is the resistivity of the solution, $\rho_c$ is the resistivity of the cell interior, $C_m$ is membrane capacitance per unit area, and R is cell radius. Assuming a $\rho_c$ of 200 Ω·cm and $C_m$ of 1.1 µF/cm, $\tau_m$ was estimated to be 66 ns for *E. coli* (R = 0.48 µm), 55 ns for *L. innocua* (R = 0.40 µm), and 135 ns for *S. enterica* (R = 0.98 µm). Cell radius was estimated by ½$\sqrt{l*w}$, where w and l are median widths and lengths from commonly listed ranges.600 ns, square-wave, monopolar pulses were delivered to 85 µL of suspension in 1 mm gap electroporation cuvettes (Bulldog Bio, Portsmouth, NH, USA) at 10 Hz. The electric field strength used in this study averaged 21 kV/cm (from after 90% rise to before 90% fall).

180 µL of cell suspension were vortexed with 20 µL of 1, 3, or 5% H$_2$O$_2$ for 15 s. Electrical exposures began 1 min after H$_2$O$_2$ addition. Single train samples were exposed to 1,000 pulses and remained in 0.1, 0.3, or 0.5% H$_2$O$_2$ for 5, 10, 15, 30, or 45 min total. For the split-dose

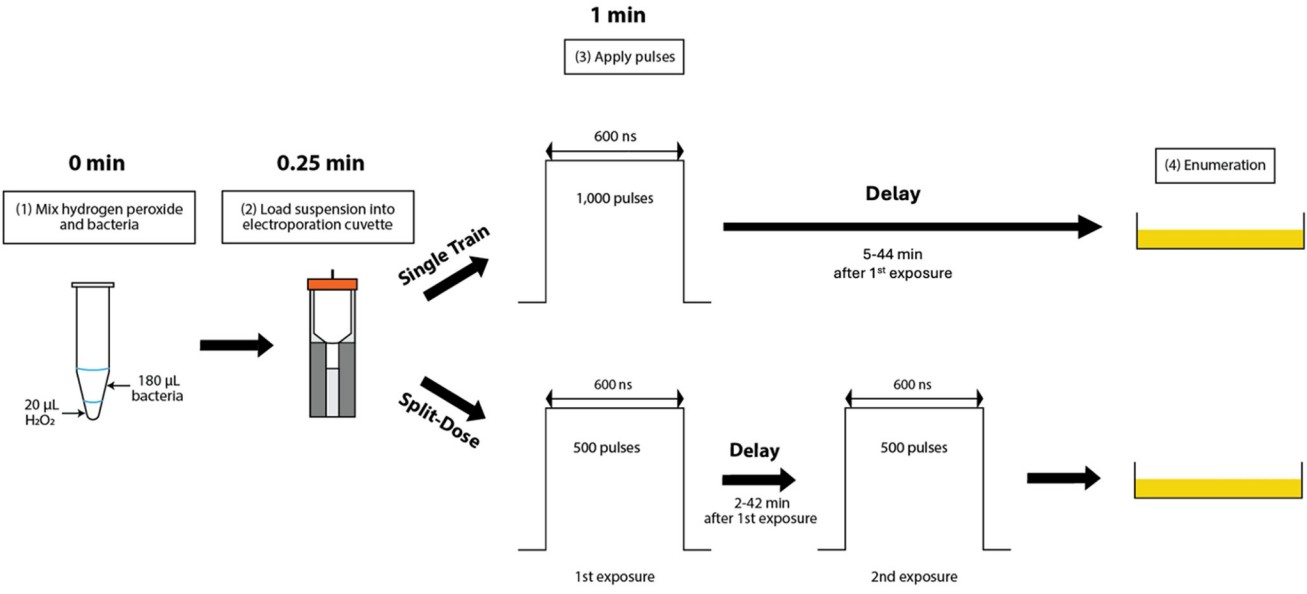

**Fig 5. Flowchart outlining the primary experimental procedures.**

treated samples, 500 pulses were delivered 1 min after H$_2$O$_2$ addition. Single trains applied 600 μs of total PEF treatment time over 100 s and split-dose were half of that. A delay of 2, 7, 12, 27, or 42 min occurred before the second 500 pulse delivery for the 5, 10, 15, 30, and 45 min contact times, respectively. A flowchart of this procedure is found in Fig 5. The total specific energy input for all exposures was calculated to be 117.6 kJ/kg.

## Electrical exposures without additive

Exposures without any H$_2$O$_2$ were also conducted using the same PEF parameters, where the maximum delay assessed matched those for the H$_2$O$_2$ treatments. A 30 min delay was also investigated for *E. coli* to further examine electrosensitization effects.

## CFU determination

Colony forming units (CFU) were determined using the track-dilution method [142]. Samples were diluted 10-fold four times in distilled water. 10 μL of each dilution, including the original sample, were spotted in ascending concentration at the top of a square petri dish containing 30 mL of LB agar. The plates were tipped to allow migration of the spots to the opposite side of the plate, forming parallel tracks. Colonies were enumerated following 16h of growth at 37˚C. *L. innocua* plates were incubated at 37˚C for 32h to provide more well-defined colonies for enumeration. The limit of detection for this method is 100 CFU/mL.

## COMSOL modeling

COMSOL Multiphysics ver. 6.0 (Burlington, MA, USA) is a finite element software that was employed to numerically determine temperature changes within the cuvette(s) using the electric current and heat transfer in solids Multiphysics coupling.

Temperature change from electric pulses is caused by Joule, or ohmic, heating, where flow of electric current creates thermal energy, and is represented by the following equation in

COMSOL:

$$\rho C_P \frac{\partial T}{\partial t} + \rho C_P u \cdot \nabla T = \nabla \cdot (k \nabla T) + Q \tag{2}$$

where ρ is density (kg/m$^3$), C$_p$ is specific heat capacity (J/kg/K), T is temperature (K), u is the displacement vector, k is thermal conductivity (W/m/K), and Q is heat load (W/m$^3$). Q can be calculated by Eqs (3)–(5):

$$Q = J \cdot E \tag{3}$$

$$J = \sigma E \tag{4}$$

$$E = -\nabla V \tag{5}$$

where J is current density (A/m$^2$), E is electric field (V/m), and V is applied voltage.

The cuvette model (S5 Fig) consisted of electrodes with properties of aluminum, plastic with properties of polystyrene, buffer with properties of water, aside from a differing electrical conductivity, and was entirely surrounded by air. Plastic was considered to be electrically, but not thermally, insulative, and air was considered to be both. The simulation was run for 500 square-wave, 21 kV/cm pulses of 600 ns duration, at a 10 Hz frequency. A finer, free tetrahedral mesh was used for the electrodes and buffer, and a coarser mesh was used elsewhere.

## Microscopy

To monitor potential clumping of the bacteria, select exposures were monitored using the Olympus CKX53 microscope (Center Valley, PA, USA) with the LCACHN-IPC 40X/0.55 NA objective. Images were captured using a PL-D752MU-T camera (Pixelink, Ottawa, ON, Canada).

## Statistical analysis

Experiments were done in triplicate, and data are expressed as mean ± standard deviation unless otherwise stated. Statistical significance was determined using GraphPad Prism ver. 10.1.1 (GraphPad Software, San Diego, California, USA).

## Supporting information

**S1 Fig. Log reductions of *E. coli*, *L. innocua*, and *S. enterica* for PEF as individual treatment.**
(PDF)

**S2 Fig. Log reductions of *E. coli*, *L. innocua*, and *S. enterica* for H$_2$O$_2$ as individual treatment.**
(PDF)

**S3 Fig. Representative images of *L. innocua* after PEF/H$_2$O$_2$ exposures.**
(PDF)

**S4 Fig. Schematic of custom electroporator.**
(PDF)

**S5 Fig. Schematic of cuvette for COMSOL simulation.**
(PDF)

**S6 Fig. Modeling of temperature effects of PEF treatment.**
(PDF)

**S7 Fig. Isobologram analysis showing combination treatment synergy.**
(PDF)

**S1 Table. Experimental determined temperature changes after PEF.**
(PDF)

**S2 Table. Statistical analysis of data: Two-way ANOVA and Tukey's multiple comparisons.**
(PDF)

## Acknowledgments

The authors thank Karl Dyer for assistance in assembling the custom electroporator and Rowan University for institutional/administrative support.

## Author Contributions

**Conceptualization:** Gary L. Thompson, Lark J. Perez.

**Data curation:** Zachary Rosenzweig, Lark J. Perez.

**Formal analysis:** Lark J. Perez.

**Funding acquisition:** Gary L. Thompson, Lark J. Perez.

**Investigation:** Zachary Rosenzweig, Jerrick Garcia.

**Methodology:** Gary L. Thompson, Lark J. Perez.

**Project administration:** Lark J. Perez.

**Resources:** Lark J. Perez.

**Supervision:** Gary L. Thompson, Lark J. Perez.

**Validation:** Zachary Rosenzweig, Lark J. Perez.

**Writing – original draft:** Zachary Rosenzweig.

**Writing – review & editing:** Gary L. Thompson, Lark J. Perez.

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
