## [Decision Letter · Decision Letter 0]

12 Aug 2024

PONE-D-24-29342Inactivation of Bacteria Using Synergistic Hydrogen Peroxide with Split-Dose Nanosecond Pulsed Electric Field ExposuresPLOS ONE

Dear Dr. Perez,

Thank you for submitting your manuscript to PLOS ONE. After careful consideration, we feel that it has merit but does not fully meet PLOS ONE’s publication criteria as it currently stands. Therefore, we invite you to submit a revised version of the manuscript that addresses the points raised during the review process.

We look forward to receiving your revised manuscript.

Kind regards,

Helen Onyeaka, PhD

Academic Editor

PLOS ONE

Journal Requirements:

"This work was funded by the USDA National Institute of Food and Agriculture's Agriculture and Food Research Initiative (NIFA AFRI) Food Safety and Defense Program (grant # 2022-67018-36540)."

4. Please note that funding information should not appear in the Acknowledgments section or other areas of your manuscript. We will only publish funding information present in the Funding Statement section of the online submission form. Please remove any funding-related text from the manuscript. 

**Additional Editor Comments:**

 I recommend that the article be accepted **after major revisions**. The study presents valuable findings on the synergistic effects of PEF and H2O2 in bacterial inactivation, which is a significant contribution to the field of food safety. However, several areas require improvement to enhance the clarity, depth, and rigor of the manuscript.

Clearly state the hypothesis of the study to guide the reader's understanding of the research objectives.

Include a flowchart or schematic diagram summarizing the experimental procedures.Provide more justification for the choice of H2O2 concentrations and contact times.

Expand on the statistical analysis, particularly the interpretation of ANOVA results.Discuss the variability in the results, especially concerning the standard deviations, to assess the reliability of the findings.

Critically evaluate the limitations of the study, including potential challenges in industrial applications.Compare the findings with other recent studies to position this research within the broader field.

Proofread the manuscript to correct minor grammatical and typographical errors.

By addressing these revisions and the revisions by the reviwers , the manuscript will be significantly strengthened and will make a substantial contribution to the literature on microbial control and food safety.

Note: HTML markup is below. Please do not edit.]

Reviewers' comments:

Reviewer's Responses to Questions

**Comments to the Author**

1. Is the manuscript technically sound, and do the data support the conclusions?

Reviewer #1: No

Reviewer #2: Yes

Reviewer #3: Yes

Reviewer #4: No

Reviewer #5: Yes

2. Has the statistical analysis been performed appropriately and rigorously? 

Reviewer #1: No

Reviewer #2: I Don't Know

Reviewer #3: Yes

Reviewer #4: No

Reviewer #5: Yes

3. Have the authors made all data underlying the findings in their manuscript fully available?

Reviewer #1: Yes

Reviewer #2: Yes

Reviewer #3: Yes

Reviewer #4: Yes

Reviewer #5: Yes

4. Is the manuscript presented in an intelligible fashion and written in standard English?

Reviewer #1: No

Reviewer #2: Yes

Reviewer #3: Yes

Reviewer #4: No

Reviewer #5: Yes

5. Review Comments to the Author

Reviewer #1: The paper reports on the combination of PEF and hydrogen peroxide to inactivate Listeria, Salmonella and E. coli in Gomori buffer. The research area is of interest although there are concerns over the structure of the script, experimental approach and interpretation of results. The Abstract could be made more informative to include the methodology and key results. The Introduction should provide more details on the state-of-the-art and basis for the study. Indeed, it was only when reading the Discussion that the terms of split-dose was actually defined. The Materials and Methods should be described in sufficient detail to allow replication. Also there is a need to describe what constitutes treatment time. From the text it appears the holding period when the hydrogen peroxide was added.

The Discussion is primarily composed of speculative statement that have no data to support the various theories. Indeed, the study reads as preliminary and observational with little attention for elucidation of mechanisms. In addition, it would have been informative to have performed trials in a food system given this was highlighted several times within the script.

Specific points

Title: Which bacteria? Which matrix?

Abstract

Line 5: In what matrix?

Line 6: What are split-dose trains? Was PEF applied for 5-45 mins?

Line 9: What is the contact time? PEF is typically a brief exposure.

Line 11: What were the log reductions for the individual treatments?

Introduction

Line 5: There are biological options.

Line 9: What about irradiation and thermal sterilization?

Line 12: Do the authors mean non-thermal methods to retain the raw quality of foods?

Line 17: Is heat generated?

Line 22: Norovirus is the number one cause of foodborne illness. Of the bacteria, Salmonella, Campylobacter and Clostridium perfringens are the most common. E. coli and Listeria are the most virulent.

Line 27: Sub-lethal treatments essentially means negligible log count reductions. The authors need to expand on the benefits of nsPEF and how these differ to normal PEF in terms of pulse time & waveform.

Line 14 and throughout: Statements should be supported by citing references.

Line 23: Unclear what is meant by split-dose.

Line 31: The authors should expand on previous work on synergistic effects. Which bacteria, which matrix and how was synergistic action demonstrated?

Line 42: Low concentrations of hydrogen peroxide followed by an antioxidant dip are typically applied. The authors should elaborate on maximum levels and allowable residue levels.

Line 44: There are codex standards for agricultural water.

Line 4: Hydrogen peroxide is relatively expensive compared to other sanitizers. It can also result in bleaching of foods.

Line 10: What was the rationale for selecting the bacteria to test? Which matrix?

Results

Table 1: Should include statistical analysis.

Line 18: Assume 3000P refers to 3999 pulses. Disinfection of what?

Line 21: Species or genera?

Line 24 and throughout: The authors should compare data in statistical terms.

Line 4: The paragraph starting “effectiveness of PEF” is confusing and unclear what point the authors are trying to make. It is unclear what single or split-dose infers.

Line 20: What log reductions are obtained? Is a 5 log CFU reduction required?

Line 18: Increased time between pulses? This should be clarified in the revised script.

Discussion

Line 10-28: This belongs in the Introduction. The authors need to discuss the results obtained in the current study.

Line 1: Was the work published? This should be described in the Introduction.

Line 6: Could the hydrogen peroxide sensitize the cells to PEF rather than vice versa?

Line 8: The authors should avoid speculating on mechanisms given no studies, apart from SEM, was performed.

Line 20: It is more likely the split dose causes inversions in the membrane potential, although it depends on the waveform.

Line 22: Catalase would degrade hydrogen peroxide rather than hydroxyl-radicals.

Line 16: Was the temperature monitored?

Line 20: What plant cells specifically?

Line 32: Speculative on applications.

Figure 5 can be removed as it doesn’t contribute to the work.

Materials and Methods

Line 7: Bacteria and cultivation conditions is a better title.

Line 8: What was the rationale for selecting the bacteria? How were the bacteria stored and revived?

Line 12: How was agitation applied?

Line 16: What is the composition of Gomori buffer?

Line 22: What did an OD 0.1 equate to in CFU/ml?

Line 23: How confident were the researchers that the hydrogen peroxide was not degraded by the cells prior to PEF treatment.

Line 28: How was the hydrogen peroxide neutralized prior to enumerating survivors?

Line 20: How long did it take to deliver 1000 pulses?

Line 34: The authors applied an unusual method for plating by spotting followed by tipping the plate. How were the bacteria evenly distributed using the method?

Reviewer #2: The manuscript "Inactivation of Bacteria Using Synergistic Hydrogen Peroxide with Split-Dose Nanosecond Pulsed Electric Field Exposures" is generally well-written and presents results that are well and extensively discussed. However, there are some aspects of the scientific language that should be addressed, as well as the presentation of results.

Page 3, line 2: “The decontamination of microbial contamination…”. There is redundancy and inaccuracy in this sentence. In fact, the decontamination is of the product itself and usually results in a decrease in the contamination…;

This error is repeated throughout the manuscript, for example on page 4 (line 27), page 5 (line 18), page 6 (lines 3 and 7), page 10 (line 29);

Page 3, line 27: “...the inactivation of other microbes like viruses…”. Viruses are biological agents but not microbes;

Page 4, lines 4-7: the sentence should be rewritten for the sake of clarity;

Figure 1: the bars in Figure 1, specifically those for treatment with hydrogen peroxide alone, do not match the values presented in Table 1;

Page 8, line 4: “The effectiveness of PEF treatments varies between the different bacteria, however, is impacted in a dose-dependent manner correlated to contact time for all bacterial species evaluated”. According to Table 1, this is not true for Salmonella (single train 0.1% 5 min and 10 min, 0.5%, 30 and 45 min);

Page 17, lines 1-3: The author claims that temperature shifts after PEF were determined using the COMSOL software. Authors should present the initial and final real temperatures (not estimated) after the several PEF treatments;

Page 13, line 39 to page 14, line 10: authors acknowledge that heating caused by electric pulses has an effect on cell death and use the COMSOL software to calculate the temperature increase. This is a fundamental issue, and the authors should clarify whether they measured temperatures before and after the PEF treatment. Additionally, it would be interesting to understand the effect of thermal treatment on the reduction of microbial load at the temperature reached after PEF, but without the use of PEF.

Reviewer #3: The authors did a good job in this manuscript. They combined nsPEF with H2O2 to kill different strains of bacteria. The results showed an enhanced effect for the combination compared to each technique separate. The introduction showed clearly the advantages of using nsPEF over micro or millisecond PEF in saving power. Description of the experimental procedures was concise and contained detailed descriptions of well-established procedures. The results fully discussed.

Reviewer #4: The manuscript presents an interesting study on the use of Pulsed Electric Fields (PEF) combined with hydrogen peroxide for microbial inactivation. While the concept has potential applications in food safety, it would be more relevant to test this method on a food matrix rather than in a buffer system. The methodology needs clearer explanation and correction in certain sections. Additionally, the results are limited and not sufficient for a scientific publication, and the discussion is largely speculative. Overall, the manuscript, in its current form, lacks sufficient data and rigorous testing to fully support its conclusions. Further details and suggestions are provided below.

Title: What bacteria were tested? Title should be more descriptive.

Abstract

Line 4: Was the bacterial reduction achieved in water, food matrix, or another medium?

Line 8: CFU per?

Line 8: Which condition resulted in more than 5 log reductions?

Line 10: ‘’Split-dose PEF exposures enhance lethality for several tested conditions, indicating greater susceptibility to PEF after oxidative damage has occurred.’’ How and why is its effect different from single dose? Do you have any evidence for your conclusions?

Key words: The selected keywords do not accurately reflect the focus of the study.

Introduction

The introduction begins by discussing food decontamination, yet the method described in the paper is not applied to food. This discrepancy should be addressed for clarity. Although the research gap is identified, it would benefit from a clearer and more detailed articulation. The stated objectives could be more specific. For instance, the rationale behind combining PEF and hydrogen peroxide needs to be better explained. It would be helpful to understand the underlying theory and the expected outcomes of using these two methods together. Clarifying these aspects would provide a stronger foundation for the study’s aims and enhance the overall coherence of the paper.

- Page 3, Line 14: "PEF in food processing?" Consider expanding on why PEF is chosen for this study and its significance in food processing. A brief overview of its advantages would be helpful.

- Page 4, Line 34: "Hydrogen peroxide role?" Provide a brief explanation of why hydrogen peroxide is used in conjunction with PEF. What is its expected role in microbial inactivation?

- Page 5, Line 8: "Herin?" The word "Herin" seems to be a typo. It should be "Herein" or another word depending on the intended meaning.

Results

Several figures labeled as supplementary are essential for a comprehensive understanding of the results, particularly Figures S1, S2, and S3. These figures are critical for elucidating the process described. However, the paper lacks sufficient data to be considered a rigorous scientific study, and many of the figures largely duplicate the information found in Table 1. Furthermore, although COMSOL modeling is discussed in the methodology section, its outcomes are not presented in the results section. To improve the paper, it is crucial to conclude which treatment combinations were most effective and to offer a brief interpretation of these findings directly within the results section. This would enhance the clarity and depth of the analysis provided.

- Page 5: What were the results for split-dose PEF treatment alone—45 min? The supplementary figures are necessary to give readers an idea of the process.

- Page 6, Line 10: ‘’No bacterial growth observed following treatment.’’ Do you mean enrichment? How do you consider it? What was the limit of detection in your trial?

- Table 1: The format of the table is not suitable for a manuscript.

- Table 1: In order to compare the data in the table, a statistical analysis is needed.

- Figure 1: What was your reason for choosing these conditions to show in the picture? If you wanted to show the best results for each treatment, for E. coli wasn’t it 0.5%–15 min? If it shows the conditions with lower concentrations of H2O2 that resulted in more than a 5 log CFU/ml reduction of bacteria, for E. coli it is 0.1%, 15 min, split-dose PEF.

- Figure 2: It is mentioned in the abstract that ‘’Split-dose PEF exposures enhance lethality for several tested conditions, indicating greater susceptibility to PEF after oxidative damage has occurred.’’ Based on the picture, this ability is only observed in the condition of 0.1% hydrogen peroxide addition to 0.3% H2O2 only for E. coli. What is the reason for better/similar results of a single dose in a higher dosage of hydrogen peroxide?

- Page 8, Line 15 to 30: What are the reasons for these results? Contact time or concentration of hydrogen peroxide?

- Page 9, Line 10: What do you conclude by observing the microscopic pictures? (I wasn’t able to find the supplemental figures).

- Page 9, Line 12 to 14: Significance of findings? the statistical analysis is needed for Table 1 to be able to discuss it. The results for single and split doses on Listeria reduction for 0.5% and 0.3% H2O2 for 30 and 45 min look similar.

- Page 9, Line 14 to 22: Why did PEF delay time have an effect on bacterial availability? What is the theory behind it? Why is Salmonella more resistant to PEF delay time compared to Listeria and E. coli?

Discussion

The discussion presents various theories and ideas from other researchers, but it lacks organization and comprehensiveness in explaining the synergistic effect between PEF and hydrogen peroxide. For example, while the authors frequently address the formation of hydroxyl radicals, it remains unclear whether these radicals are produced when PEF is combined with hydrogen peroxide. Clarifying this potential formation could significantly enhance the interpretation of the results. Additionally, the discussion does not provide sufficient evidence from the results to support its speculative claims. To strengthen the paper, it would be beneficial to compare the effectiveness of the PEF-hydrogen peroxide combination with other microbial inactivation methods, providing valuable context. Furthermore, the paper should address any study limitations and propose directions for future research, especially regarding the scalability of PEF-hydrogen peroxide treatment in industrial applications.

- Page 10, Line 24 to 28: "With a compatible electrode material and a buffer consisting of no added transition metal ions, the H2O2 reduction must be catalyzed by bioavailable ions. It is estimated that a third of all proteins contain metal and 40% of enzymes with known structures are metal-dependent. While most of the total metal content within a bacterium cell is bounded, free metal ions also exist on the μM scale. H2O2 will specifically oxidize enzymes and proteins bonded to iron, in addition to Fe2+ ions within the labile pool [85]. Pore formation from PEF will cause either the efflux of intracellular contents or the influx of H2O2, exposing the chemical to a much higher concentration of metal ions compared with diffusion. The quicker formation of hydroxyl radicals will generate more significant damage to the surrounding proteins, enzymes, lipids, and DNA.’’ Based on this theory, is it better to use PEF and hydrogen peroxide at the same time or expose the bacteria to PEF first and then add hydrogen peroxide?

- Page 11, Line 13, 14: ‘’In a general sense, we hypothesize that oxidative damage to the membrane structure induces greater susceptibility to PEF.’’ How do you hypothesize the damage in the membrane without microscopy consideration?

- Page 12, Line 2: Which evidence suggests oxidative damage aids in enhancing susceptibility to PEF exposures? Only increasing the log reduction level?

Materials and Methods

- Page 15, Line 5: "Initial microbial levels?" Mention the initial levels of microorganisms and the detection limits used in the study. This is crucial for interpreting the effectiveness of the treatments.

- Page 15, Line 15: "Buffer choice?" Explain the rationale behind the choice of buffer, pH, and the concentration range of hydrogen peroxide. How do these factors influence the results?

Page 15, Line 30, a Schematic picture of the unit will help the reader understand the process better

Page 16, line 35, is 16 h enough for growing Listeria, E.coli and Salmonella on LB?

Page 17, line 24, wasn’t better to use an electron microscope to consider the membrane damage instead of a light microscope?

Reviewer #5: The paper aims to fill a gap in the literature for nsEP application as a means of bacterial disinfection. Specifically, the authors note that nsEP alone and nsEP plus antibiotics have been investigated to some success as well as split dose approaches to increase inherent sensitivity of the system to nsEP but neither in combination. They therefore focus on assessing the combined effects of split dose nsEP and hydrogen peroxide. The authors introduce H202 as a optimal means of sterilization across many applications due to its relatively non-toxic nature, disposal via dissociation, low cost, and lack of smell/taste. The authors did well to expose three unique bacteria types to nsep, h202, and combined exposures. Overall, they found that all bacteria were impact by the combined exposure and split exposure impacted all bacteria (although the effect was not universal or straightforward). The authors did a great job of controlling exposure time across all samples and concentrations of h202. This is admirable as I found myself asking questions about consistency in incubation time of the cells per exposure paradigm and found they had noted in the methods that they had held all samples for the same amount of time under each condition. This is rarely done in papers and shows a high level of control and quality of the experiments performed. Additionally, the results and discussion sections are thorough and leave little unturned as possible mechanism(s) for the results or future studies to do. The author does mention that oxidation of phospholipids has been both shown in impact nsEP sensitivity and also be driven by nsEP exposure (in a couple papers). This is a key point as the point of action of H202 in these bacteria is not identified by the author nor its dependence on time well explored. Understandably this is beyond the goal of this paper, but it is key to determining if the mechanism is truly functioning on amplification of the same mechanism or an interplay between two stressed that presents as reduction in bacterial population. The key drawback of the paper is therefore the lack of a mechanism of action outside of speculation in how these two stressors are working together on the bacteria. it is noted that the mechanism for nsEP remains unknown for cells both mammalian and bacteria, so it would be difficult for the authors do anything more than speculate and maintain the focus of this paper on the synergetic effects of nsEP and h020 exposures.

The paper was well written and I have no negative comments regarding its quality. The paper was thoroughly reviewed by the authors and the figures were made clearly and concisely. I do question how applicable nsEP and H202 could be in a commercial application, but given this papers results, you would definitely need less of both to achieve a similar outcome of either alone. This would drive down costs of pulser systems which are currently being deployed in clinical waste water applications. This paper defined a gap in knowledge regarding nsEP for bacterial decontamination and thoroughy filled it beyond expectations by studying a large range of parameters across three dissimilar species. The authors also noted future work that needs to be done to better refine and describe the results. I believe this paper should be accepted for publication.

Minor Points:

Some of the supplemental figures could easily be in the paper as the trends with increasing dose are better understood by graph as opposed to table, especially unexpected trends.

Page 8 Line 4 “..however is impacted” reads odd.

6. PLOS authors have the option to publish the peer review history of their article (what does this mean?). If published, this will include your full peer review and any attached files.

Reviewer #1: **Yes: **Keith Warriner

Reviewer #2: No

Reviewer #3: **Yes: **Amr Abd-Elghany

Reviewer #4: No

Reviewer #5: No

---

## [Author Response · Author response to Decision Letter 0]

12 Sep 2024

We acknowledge the reviewers for their thoughtful feedback on our manuscript. We have made changes throughout the document in response to the comments which is described in the attached response to reviewers file.

---

## [Editor Report · Decision Letter 1]

17 Sep 2024

Inactivation of Bacteria Using Synergistic Hydrogen Peroxide with Split-Dose Nanosecond Pulsed Electric Field Exposures

PONE-D-24-29342R1

Dear Dr. Perez,

We’re pleased to inform you that your manuscript has been judged scientifically suitable for publication and will be formally accepted for publication once it meets all outstanding technical requirements.

Kind regards,

Helen Onyeaka, PhD

Academic Editor

PLOS ONE
---

## [Editor Report · Acceptance letter]

25 Sep 2024

PONE-D-24-29342R1 

PLOS ONE

Dear Dr. Perez, 

I'm pleased to inform you that your manuscript has been deemed suitable for publication in PLOS ONE. Congratulations! Your manuscript is now being handed over to our production team.

Kind regards, 

on behalf of

Dr. Helen Onyeaka 

Academic Editor

PLOS ONE